# Effect of External Fields on the Electronic and Optical Properties in ZnTe/CdSe and CdSe/ZnTe Spherical Quantum Dots

Rafael G. Toscano-Negrette [1,2,*], José C. León-González [1,2], Juan A. Vinasco [1], Alvaro L. Morales [1,*], Miguel E. Mora-Ramos [3] and Carlos A. Duque [1]

1   Grupo de Materia Condensada-UdeA, Instituto de Física, Facultad de Ciencias Exactas y Naturales, Universidad de Antioquia UdeA, Calle 70 No. 52-21, Medellín 050010, Colombia; jose.leong@udea.edu.co (J.C.L.-G.); juan.vinascos@udea.edu.co (J.A.V.); carlos.duque1@udea.edu.co (C.A.D.)
2   Departamento de Física y Electrónica, Universidad de Córdoba, Carrera 6 No. 77-305, Montería 230002, Colombia
3   Centro de Investigación en Ciencias, Instituto de Investigación en Ciencias Básicas y Aplicadas, Universidad Autónoma del Estado de Morelos, Av. Universidad 1001, Cuernavaca CP 62209, Morelos, Mexico; memora@uaem.mx
*   Correspondence: rafael.toscano@udea.edu.co (R.G.T.-N.); alvaro.morales@udea.edu.co (A.L.M.)

**Abstract:** A theoretical analysis was conducted to examine the electronic and optical properties of a confined electron and a hole in a type-II core-shell spherical quantum dot composed of CdSe/ZnTe and ZnTe/CdSe. The Schrödinger equation for the electron and the hole was numerically solved using COMSOL-Multiphysics software in the 2D axisymmetric module, which employs the finite element method under the effective mass approximation. A Fortran code was utilized to calculate excitonic energy, specifically designed to solve the Coulomb integral. The calculations encompassed variations in the inner radius ($R_1$), as well as variations in the electric ($F_z$) and magnetic ($B$) fields along the $z$-axis. The absorption coefficients were determined for transitions between the hole and electron ground states, considering $z$-polarized incident radiation. Including a magnetic field increases the transition energy, consequently causing the absorption peaks to shift toward the blue region of the spectrum. On the other hand, the electric field decreased the overlap of the electron and hole wavefunctions. As a result, the amplitude of the absorption peaks decreased with an increase in the electric field.

**Keywords:** type-II quantum dot; absorption coefficient; ZnTe/CdSe; CdSe/ZnTe; magnetic field; electric field





## 1. Introduction

There is an increasing demand for electronic devices based on semiconductor heterostructures at the nanometer scale. This demand has led to a rise in theoretical and experimental studies to discover new properties (such as electronic, optical, and optoelectronic) or improve existing ones in these devices.

When two semiconductor materials with different forbidden energy gaps are brought together, one acts as the potential well region while the other acts as the barrier region for electrons and holes. A quantum well is formed when the motion of electrons and holes is confined in one dimension. Similarly, confinement in two spatial dimensions results in a quantum wire, and confinement in all three dimensions leads to quantum dots (QDs), where the whole energy spectrum of particles becomes discrete. These systems are highly sensitive to geometry, size, and external probes, such as magnetic and electric fields [1–4]. They are also influenced by the shallow-donor and -acceptor impurities [5–7], neutral and charged excitons [8–10], intense resonant and non-resonant lasers [11], hydrostatic pressure, temperature [12], and the inclusion of spin–orbit and Zeeman effects, represented by the Rashba and Dresselhaus terms [13,14]. All these effects significantly modify the electronic,

optical, thermal, and mechanical properties of semiconductor nanostructures. It is worth recalling that when considering most of the external effects mentioned earlier, analytical solutions for the problem of finding energy eigenstates are not available. Therefore, various numerical methods are employed, such as the finite element method (FEM) and the finite difference method.

Moreover, QDs can be fabricated in the experimental domain using molecular beam epitaxy (MBE), hydrothermal method, and ultrasonic sol-gel. These techniques allow for precise control over the dimensions and composition of the QDs [15–20]. The semiconductor properties of QDs are highly influenced by their shapes and sizes compared to the constituent materials [21]. Among the various 2D geometries studied, we have disks and rings, while in the 3D realm, there are structures like cylinders, cones, pyramids, and spherical shapes [22–26]. The primary focus of this work is to investigate type-II core-shell QDs with a spherical shape. These QDs are characterized by having the conduction band minimum (electron confinement) spatially separated from the valence band maximum (hole confinement). This implies that when electrons are confined in the core, the holes are confined in the shell, or vice versa.

An example of a type-II QD is formed by combining two semiconductor materials, CdSe and ZnTe. In this case, CdSe acts as the hole barrier and electron well, while ZnTe is the hole well and electron barrier. The literature contains numerous reports emphasizing the excellent electronic and optical properties of type-II QDs based on ZnTe and CdSe [27–29]. One notable study was conducted by Chen et al. where they synthesized and characterized CdSe/CdTe/ZnTe type-II core-shell-shell QDs. They discovered an extraordinarily long radiative lifetime of 10 ms for interband emission due to the spatial separation of electrons and holes between CdSe and ZnTe facilitated by the CdTe intermediate layer [30]. In 2023, Jaouane et al. performed a theoretical analysis of the optical properties of spherical core-shell QDs with a modified Kratzer potential. Their investigation focused on transitions between the ground and the first excited states. They observed that the absorption coefficients were redshifted by varying the parameter $r_e$, representing the potential bond length. Additionally, they found that these quantities were blue-shifted when shielding and control parameters of the Kratzer potential were varied [31]. Another study in 2023 by Koç et al. explored the electronic and optical properties of excitons and biexcitons in multi-shell CdS/ZnSe/ZnTe/CdSe QDs as a function of the core size. The researchers reported that the size of the ZnSe shell could be manipulated to control the overlap between electron and hole wavefunctions while keeping the exciton and biexciton transition energy nearly unchanged. This ability to engineer the wavefunctions in type-II multilayers allowed for a wide range of tunability in radiation lifetime without significant changes in transition energies [32]. Restrepo and co-authors developed a theoretical model to describe excitonic states in colloidal spherical QDs with quasi-infinite confinement potential [33]. Using a two-band model, they found an excellent match between their theoretical findings and the experimentally reported photoluminescence energy, obviously, with minor differences.

Type II quantum dots (QDs) undergo significant changes in their electronic and optical properties when exposed to external fields. In a study conducted by Holovatsky et al. in 2023 [34], they theoretically explored the effects of a magnetic field on the electronic properties and optical quantum interband transitions in type II ZnTe/CdSe and CdSe/ZnTe QDs. Their findings revealed that the magnetic field breaks the spherical symmetry of the system, resulting in energy spectrum degeneracy concerning the magnetic quantum number. Moreover, they observed a dependence of the oscillator strength on the magnetic field, as the field affects the overlap of electron and hole wavefunctions. In 2007, Kuskovsky et al. [35] presented experimental and theoretical studies on magnetoexcitons in type II QDs formed in Zn-Se-Te multilayers. Their primary objective was to investigate the impact of an intense magnetic field (31,T) on the optical properties. The results demonstrated that the magneto-photoluminescence exhibited a non-monotonic behavior of the exciton emission intensity concerning the magnetic field, indicating the manifestation of the Aharonov–Bohm effect. These findings were further supported by the performed numerical calculations. Addition-

ally, Roy et al. in 2012 [36] conducted an experimental study on the influence of magnetic and electric fields on type II ZnTe/ZnSe QDs. The authors reported observing robust and narrow Aharonov–Bohm oscillations in the magneto photoluminescence intensity of the stacked QDs due to an embedded electric field. Furthermore, they noticed a decrease in Aharonov–Bohm oscillations, which they attributed to the electric field's presence.

Building upon the previously reported works mentioned above, this study aims to investigate the effects induced by changes in the core size and the inclusion of external electric and magnetic fields on the electronic and optical properties of type-II quantum dots (QDs) made of ZnTe/CdSe and CdSe/ZnTe. To achieve this, we will conduct numerical calculations using the Finite Element Method (FEM) within COMSOL-Multiphysics software, employing the 2D axis-symmetric module while considering the effective mass approximation. The theoretical calculations will focus on determining the energies and wavefunctions of spherical core-shell type-II QDs composed of ZnTe/CdSe and CdSe/ZnTe. These calculations will take into account variations in the internal radius ($R_1$) as well as changes in the external electric ($F_z$) and magnetic ($B$) field strengths, both aligned along the $z$-axis. To account for the Coulomb interaction between the electron and the hole, a Fortran code will be utilized. Subsequently, using the energies and wavefunctions obtained for the electron-hole system, we will examine the optical response of the QDs to light absorption by analyzing the absorption coefficient at the quantum interband transitions between the hole's ground state and the electron's ground state. The structure of the article is organized as follows: Section 2 presents the theoretical framework used in the calculations, Section 3 discusses the results obtained from the simulations, and Section 4 provides a summary of the main conclusions drawn from this study.

## 2. Theoretical Model

Figure 1 shows the pictorial views of the two investigated systems, which are type-II spherical ZnTe/CdSe and CdSe/ZnTe core/shell QDs. The inner and outer radius dimensions are $R_1$ and $R_2$, respectively. In the lowest part of the figure, we have represented a schematic view of the $r$-dependent confinement potential for valence and conduction bands. This function is set for electrons as zero within the CdSe material, $V_e$ in the ZnTe material, and infinite in the vacuum external region. This function is set for holes as zero within the ZnTe material, $V_h$ in the CdSe material, and infinite in the vacuum external region. An axially applied magnetic ($B\,\hat{k}$) and electric ($F_z\,\hat{k}$) field have been taken into account.

In Cartesian coordinates, the effective mass Schrödinger equation for an electron (hole) confined in the structure, with all the above-mentioned contributions, is written in the form:

$$\left\{ \left( \vec{p}_i - q_i\,\vec{A}_i \right) \cdot \left[ \frac{1}{2\,m_i^*(\vec{r}_i)} \left( \vec{p}_i - q_i\,\vec{A}_i \right) \right] - q_i\,\vec{F} \cdot \vec{r}_i + V_i(\vec{r}_i) \right\} \psi_i(\vec{r}_i) = E_i\,\psi_i(\vec{r}_i), \quad (1)$$

where $i = e, h$ corresponds to electron (hole), $m_i^*(\vec{r}_i)$ is the position-dependent electron (hole) effective mass, $\vec{p}_i = -i\,\hbar\,\vec{\nabla}_i$, $\vec{r}_i = x_i\,\hat{i} + y_i\,\hat{j} + z_i\,\hat{k}$, $q_e = -e$ is the electron charge, $q_h = +e$ is the hole charge, and $e$ is the absolute value of the elementary charge. Additionally, $\vec{A}_i = -\frac{B}{2}\,(y_i\,\hat{i} - x_i\,\hat{j})$ is the vector potential associated with the applied magnetic field, where $\vec{B} = \vec{\nabla}_i \times \vec{A}_i$ and $\vec{\nabla}_i \cdot \vec{A}_i = 0$.

Expanding Equation (1), and considering that the electric field is applied along the $z$-direction, we obtain:

$$\left\{ -\frac{\hbar^2}{2} \vec{\nabla}_i \cdot \left( \frac{1}{m_i^*(\vec{r}_i)} \vec{\nabla}_i \right) - \frac{i\,\hbar\,q_i\,B}{2\,m_i^*(\vec{r}_i)} \left( y\frac{\partial}{\partial x} - x\frac{\partial}{\partial y} \right) - \frac{i\,\hbar\,q_i\,B}{4} \vec{\nabla}\left( \frac{1}{m_i^*(\vec{r}_i)} \right) \cdot (y_i\,\hat{i} - x_i\,\hat{j}) \right.$$

$$\left. + \frac{q_i^2\,B^2(x_i^2 + y_i^2)}{8\,m_i^*(\vec{r}_i)} - q_i F_z z_i + V_i(\vec{r}_i) \right\} \psi_i(\vec{r}_i) = E_i\,\psi_i(\vec{r}_i). \quad (2)$$

In the absence of electric and magnetic fields, our problem has spherical symmetry, and the 3D wavefunction can be written as the product of a one-dimensional function for

the coordinate $r$ and a two-dimensional function depending on the angular coordinates, $\theta$ and $\varphi$, known as spherical harmonics. By turning on either or both electric or magnetic fields applied along the $z$-direction, the system loses spherical symmetry but preserves cylindrical symmetry about the $z$-axis. This situation is also valid when both fields are zero. In that sense, it is useful to work in cylindrical coordinates to use such symmetry. Using the azimuthal symmetry condition, it is possible to propose, in cylindrical coordinates, a solution of the type

$$\psi_i(\vec{r}_i) = \psi_i(\rho_i, \varphi_i, z_i) = R_i(\rho_i, z_i)\, e^{i\, m_i\, \varphi_i}\,, \tag{3}$$

where the $m_i$-integer is the magnetic quantum number ($m_i = 0, \pm 1, \pm 2, \dots$). Consequently, the $R_i(\rho_i, z_i)$ function satisfies the differential equation

$$\left[ -\vec{\nabla}_{2D,i} \cdot \left( \frac{\hbar^2}{2\, m_i^*(\rho_i, z_i)} \vec{\nabla}_{2D,i} \right) + \frac{m_i^2\, \hbar^2}{2\, m_i^*(\rho_i, z_i)\, \rho_i^2} - \frac{\hbar\, q_i\, B\, m_i}{2\, m_i^*(\rho_i, z_i)} + \frac{q_i^2\, B^2\, \rho_i^2}{8\, m_i^*(\rho_i, z_i)} - q_i\, F_z\, z_i + V_i(\rho_i, z_i) \right] R_i(\rho_i, z_i) = E_i\, R_i(\rho_i, z_i), \tag{4}$$

where $\vec{\nabla}_{2D,i}$ is the $\rho_i$- and $z_i$-dependent two-dimensional gradient operator.

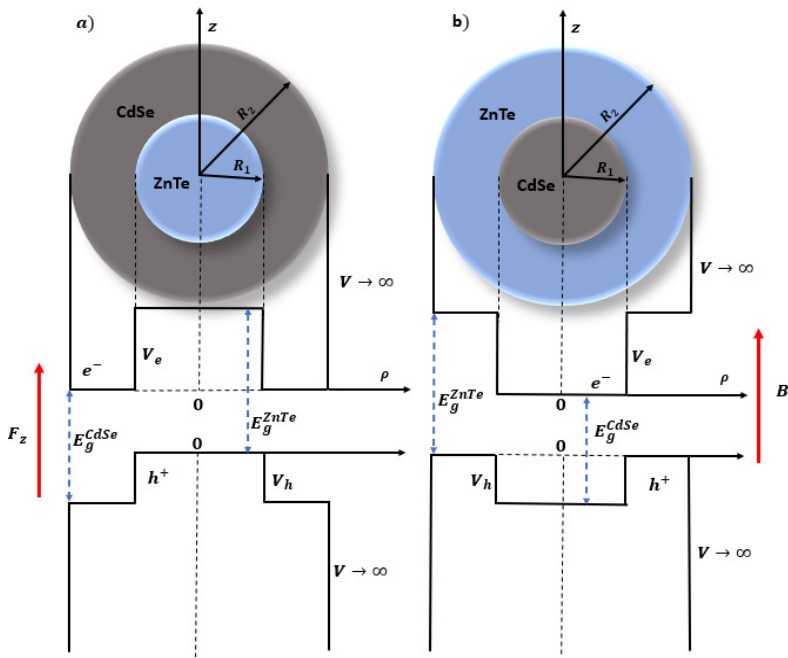

**Figure 1.** ZnTe/CdSe (**a**) and CdSe/ZnTe (**b**) spherical quantum dot cross-section. The dimensions of the QD and the axially applied electric and magnetic field are indicated together with the $r$-dependent confinement potentials.

We consider the parabolic band approximation, together with Dirichlet boundary conditions at the outer edges of the barrier matrix and the BenDaniel–Duke conditions at the inner QD interfaces (see Figure 1). Accordingly, $m_i^*$ and $V_i(\rho_i, z_i)$ both depend on the position in the heterostructure, and they are expressed for ZnTe/CdSe as

$$m_i^{*c}(\rho_i, z_i) = \begin{cases} m_h^{*\mathrm{ZnTe}} & \text{if } 0 \leq r_i \leq R_1 \\ m_e^{*\mathrm{CdSe}} & \text{if } R_1 < r_i \leq R_2 \end{cases} \tag{5}$$

and

$$V_e(\rho_e, z_e) = \begin{cases} V_e & \text{if } 0 < r_e \leq R_1 \\ 0 & \text{if } R_1 < r_e \leq R_2 \\ \infty & \text{if } r_e > R_2 \end{cases} \qquad V_h(\rho_h, z_h) = \begin{cases} 0 & \text{if } 0 < r_h \leq R_1 \\ V_h & \text{if } R_1 < r_h \leq R_2 \\ \infty & \text{if } r_h > R_2\,. \end{cases} \tag{6}$$

In the case of CdSe/ZnTe, the expressions are:

$$m_i^{*c}(\rho_i, z_i) = \begin{cases} m_e^{*\text{CdSe}} & \text{if } 0 \le r_i \le R_1 \\ m_h^{*\text{ZnTe}} & \text{if } R_1 < r_i \le R_2 \end{cases} \tag{7}$$

and

$$V_e(\rho_e, z_e) = \begin{cases} 0 & \text{if } 0 < r_e \le R_1 \\ V_e & \text{if } R_1 < r_e \le R_2 \\ \infty & \text{if } r_e > R_2 \end{cases} \qquad V_h(\rho_h, z_h) = \begin{cases} V_h & \text{if } 0 < r_h \le R_1 \\ 0 & \text{if } R_1 < r_h \le R_2 \\ \infty & \text{if } r_h > R_2 \, . \end{cases} \tag{8}$$

In Equations (5)–(8), $r_i = \sqrt{\rho_i^2 + z_i^2}$.

By solving Equation (4), the ground state wavefunctions of the electron ($\psi_e^1(\vec{r_e})$) and the hole ($\psi_h^1(\vec{r_h})$) are obtained. With them, it is possible to use a first-order perturbative approximation to evaluate the excitonic interaction by means of the following Coulomb integral [37,38]

$$E_{exc} = \frac{e^2}{4\pi\epsilon_0\epsilon_r} \int_{\Omega_h} \int_{\Omega_e} \frac{|\psi_e^1(\vec{r_e})|^2 |\psi_h^1(\vec{r_h})|^2}{|\vec{r_e} - \vec{r_h}|} dV_e \, dV_h \, , \tag{9}$$

where $\epsilon_0$ and $\epsilon_r$ are the vacuum permittivity and dielectric constant, respectively, and $|\vec{r_e} - \vec{r_h}|$ is the $e - h$ distance. Moreover, $dV_e$ and $dV_h$ represent the differential volume for electron and hole. Since the interest is focused on the magnitude of the Coulomb interaction, the negative sign of the electrostatic energy has been omitted. We want to highlight that Heyn and co-workers have used this perturbative approach in multiple studies where excitonic states in heterostructures with axial symmetry have been described and found very good agreement with experimental results for the photoluminescence peak energy transition [37,38].

Taking advantage of the azimuthal symmetry of the problem, it is possible to calculate the angular integration of Equation (9) analytically with a first-order elliptic integral ($K(x)$). This will reduce the dimensions of the integral, going from six variables, three for the electron and three for the hole ($\rho_{e(h)}, \varphi_{e(h)}, z_{e(h)}$), to four variables ($\rho_{e(h)}, z_{e(h)}$):

$$E_{exc} = \frac{\pi e^2}{\epsilon_0 \epsilon_r} \int_{-R_2}^{+R_2} \int_0^{+R_2} \int_{-R_2}^{+R_2} \int_0^{+R_2} |R_e^1(\rho_e, z_e)|^2 |R_h^1(\rho_h, z_h)|^2 \left[ \frac{8\pi K\left(\frac{r_p}{1+r_p}\right)}{r\sqrt{1+r_p}} \right] \rho_e \rho_h \, d\rho_e \, dz_e \, d\rho_h \, dz_h \, , \tag{10}$$

where $r_p = 4\rho_e\rho_h/r$ and $r = \sqrt{(\rho_e - \rho_h)^2 + (z_e - z_h)^2}$.

This study focuses on investigating the effects of external magnetic and electric fields as well as variations in the core size (represented by the parameter $R_1$), on type-II spherical QDs composed of ZnTe/CdSe and CdSe/ZnTe. The study specifically examines the confinement of holes, electrons, and excitons within these QDs. Notably, despite utilizing an external magnetic field, the Zeeman effect has been disregarded to facilitate a more detailed analysis of the energy levels of the particles under investigation because for 30 T the Zeeman effect is of the order of 0.5 meV (not shown here); this is very small compared to the orbital splitting in our results.

Equation (4) is solved using the FEM within the axis-symmetric module of the COMSOL-Multiphysics licensed software [39–41]. This solution provides the electron and hole wavefunctions and their corresponding energies. The computational settings employed in this work include a user-controlled extra fine mesh with 15,520 vertices, 30,586 triangles, 664 edge elements, and seven vertex elements. The minimum element quality is set at 0.4766, while the mean element quality is 0.9235. The area ratio element size is 0.0442, with a total mesh area of 353.4 nm$^2$. The hardware utilized consists of a single 11th-generation i7 processor with 16 GB of RAM. To numerically calculate Equation (10), a Fortran 77 code is employed. The ground state wavefunctions ($R_e^1$ and $R_h^1$), obtained

from COMSOL-Multiphysics regarding the coordinates $(\rho, z)$, are exported and used as input. By discretizing the electron and hole coordinates over the same set of mesh points, the wavefunctions are represented at these specific locations. Consequently, the integral in Equation (10) can be approximated using a Riemann sum [42].

The obtained electronic states for the electron and the hole are then utilized to evaluate the absorption coefficient, which is dependent on the transitions occurring between the ground state of the electron and the hole. The following relationship provides the expression for evaluating the absorption coefficients:

$$\alpha(\hbar\,\omega) = \frac{e^2\,\omega\,\pi}{n_r\,\epsilon_0\,c\,L}\sum_k f_k\,\delta_k(E_{e-h} - \hbar\,\omega), \tag{11}$$

where $E_{e-h}$ is the transition energy of the exciton, $\hbar\,\omega$ is the energy of the incident photon, $\delta_k$ is the line shape function, $f_k$ is the oscillator strength, and $L$ is the diameter of QD [11,26,43]. The sum is performed over all possible transitions in the system. Only the transition $1s - 1s$ of the electron and hole is considered for the absorption edge. The following relationship calculates the oscillator strength for these optical transitions:

$$f_{1-1} = \frac{2\,\pi^2\,E_p}{E_{e-h}}\left|\int_{-R_2}^{+R_2}\int_0^{+R_2} R_e^1(\rho, z)\,R_h^1(\rho, z)\,\rho\,d\rho\,dz\right|^2, \tag{12}$$

where the squared term, within the absolute value, is the overlap between the electron and hole wavefunctions [44], and $E_p$ is the Kane energy (19.1 eV for ZnTe [45] and 21 eV for CdSe [44]). A homogeneous line broadening was considered; for that purpose, a hyperbolic secant type function is selected to represent the $\delta$-function in Equation (9), i.e.,

$$\delta_{1-1}(E_{e-h} - \hbar\,\omega) \cong \frac{1}{\pi\,\hbar\,\Gamma}\,\mathrm{sech}\left(\frac{\hbar\omega - E_{e-h}}{\hbar\,\Gamma}\right). \tag{13}$$

Here, $\hbar\,\Gamma$ is the linear factor of the broadening, which is related to the lifetime, and is taken to be 30 meV [26,44]. Note that Equation (13) represents a $\delta$-function shifted so that its maximum position coincides with the $e - h$ energy transition. Finally, the transition energy is given by

$$E_{e-h} = E_g + E_e^1 + E_h^1 - E_{exc}^1. \tag{14}$$

Additionally, $E_g$ is the effective energy gap between the lower part of the CdSe conducting band and the upper part of the ZnTe valence band. For ZnTe/CdSe, $E_g = E_g^{ZnTe} - V_e$, whereas for CdSe/ZnTe, $E_g = E_g^{CdSe} - V_h$ [29].

## 3. Results and Discussion

The parameters used in this work are: $m_e^{*CdSe} = 0.13\,m_0$, $m_h^{*CdSe} = 0.45\,m_0$, $\varepsilon^{CdSe} = 10.6$, $E_g^{CdSe} = 1.75\,\mathrm{eV}$, $m_e^{*ZnTe} = 0.15\,m_0$, $m_h^{*ZnTe} = 0.2\,m_0$, $\varepsilon^{ZnTe} = 9.7$, $E_g^{ZnTe} = 2.2\,\mathrm{eV}$, $V_e = 1.27\,\mathrm{eV}$, and $V_h = 0.84\,\mathrm{eV}$ [29]. Here, $m_0$ is the free electron mass.

Figure 2 illustrates the energy variation concerning the inner radius for the ZnTe/CdSe (Figure 2a,c) and CdSe/ZnTe (Figure 2b,d) QDs. In Figure 2a–d, the electron/hole energies are depicted. Figure 2a,d demonstrate the confinement of both electrons and holes within the shell, where an increment in the inner radius leads to a reduction in the size of the shell, with the subsequent increase in energies. Conversely, Figure 2b,c display the case of confinement within the core, with energy levels decreasing as $R_1$ becomes larger. The observed behavior can be explained by considering the Heisenberg uncertainty principle. As the shell size decreases, the confinement volume for the particles diminishes, leading to an increase in energy. On the other hand, when the particle localizes within the core, the confinement volume increases with the growth of $R_1$, resulting in a decrease in energy. The results presented in Figure 2 correspond to different values of the $m$-quantum number. The first state, with $m = 0$, exhibits no degeneracy. The first excited state is triply degen-

erate, characterized by $m = 0, \pm1$, while the second excited state is five-fold degenerate, represented by $m = 0, \pm1, \pm2$. These trends align with the well-known behavior observed in a bulk hydrogen atom.

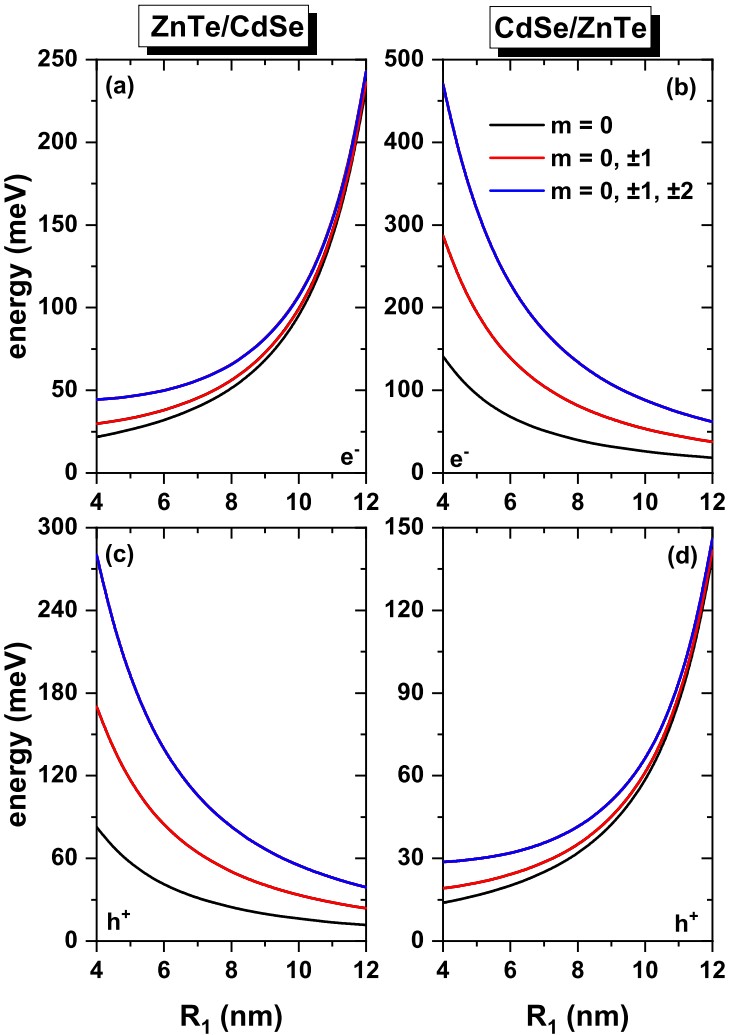

**Figure 2.** Electronic states of a spherical core/shell quantum dot of ZnTe/CdSe (**a**,**c**) and CdSe/ZnTe (**b**,**d**) as functions of the inner radius, $R_1$. Panels (**a**–**d**) present the results for electrons/holes, with $R_2 = 15$ nm, for different values of the $m$-quantum number, and without external electric and magnetic field effects.

Figure 3 demonstrates the impact of introducing a magnetic field, which leads to the breaking of degeneracy among the excited states. For instance, in the absence of a magnetic field, the initially triply degenerate first excited state now splits into three distinct states with $m = 0, \pm1$ as the magnetic field strength ($B$) increases. A similar behavior is observed for the second excited state, where introducing a magnetic field gives rise to five states. associated with $m = 0, \pm1, \pm2$. It is important to note that, for electrons, the energies corresponding to positive values of $m$ consistently increase with the magnetic field, while the energies for negative values of $m$ initially exhibit a decreasing trend until reaching a minimum value, after which they begin to rise. Conversely, for holes, the energies display an increasing trend for negative values of $m$, whereas they demonstrate a decreasing and subsequently increasing pattern for positive values of $m$. This behavior arises from the interplay between the linear and quadratic terms associated with the applied magnetic field in Equation (6). Specifically in Figure 3a,d, when the magnetic field is below 10 T, the linear term dominates, while for fields exceeding 10 T, the quadratic term becomes predominant.

This behavior gives rise to oscillations in the ground state. For $B \leq 10\,\text{T}$ the ground state corresponds to $m = 0$ (*s*-like symmetry). Within the magnetic field range of 10 T to 20 T, the ground state corresponds to $m = -1$. Subsequently, the ground state transitions to that with $m = -2$, within the range of 20 T to 30 T. This behavior holds for electrons, Figure 3a. A similar behavior is observed for holes but for positive values of $m$ within roughly the same magnetic field range, Figure 3d. These effects are more pronounced when the electron or hole is confined to the shell region, Figure 3a,d. Notably, when the electron or hole is confined to the core, the ground state with $m = 0$ retains the symmetry of an *s*-type state within the reported range of magnetic fields.

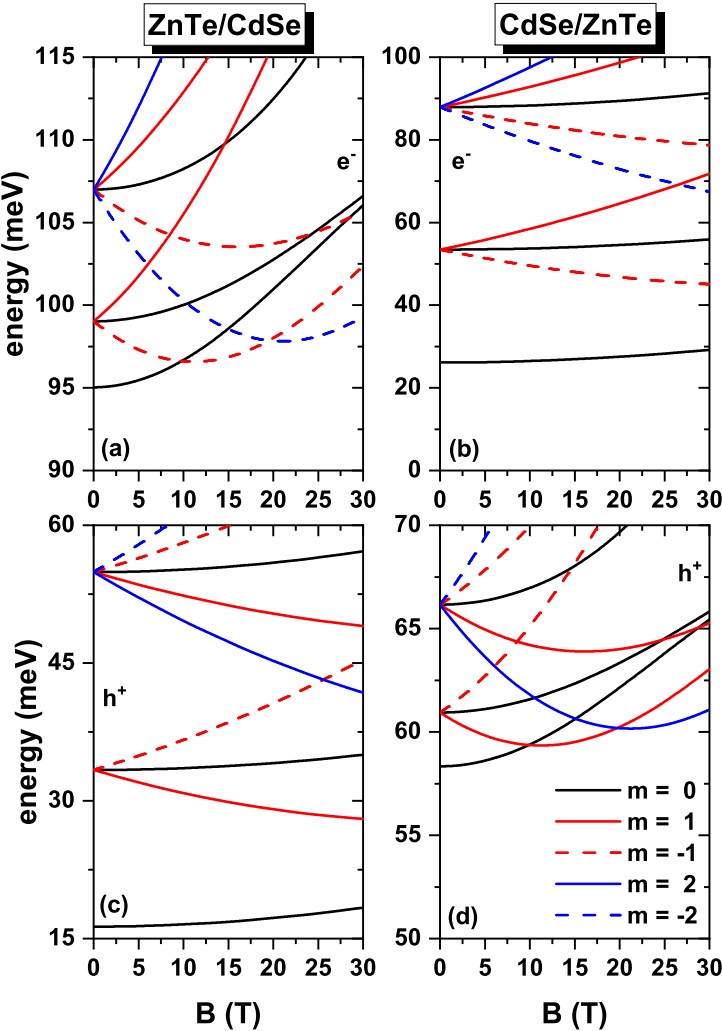

**Figure 3.** Electronic states of a spherical core/shell quantum dot of ZnTe/CdSe (**a**,**c**) and CdSe/ZnTe (**b**,**d**) as functions of the applied magnetic field, *B*. Panels (**a**–**d**) present the results for electrons/holes with $R_1 = 10\,\text{nm}$ and $R_2 = 15\,\text{nm}$, for different values of the *m*-quantum number, and without electric field effects.

Figure 4 illustrates the energy dependence on the electric field without considering the effects of an external magnetic field. The plots reveal that the electric field also disrupts the degeneracy of the excited states, resulting in two energy levels for the first excited state: one for $m = 0$ and another doubly degenerate for $m = \pm 1$. Additionally, the energy values decrease as the electric field intensity increases. This behavior is particularly pronounced when the electron or hole is confined within the shell (Figure 4a,d). Conversely, when confinement occurs in the core (Figure 4b,c), the energies do not exhibit such a significant variation in the presence of the electric field. This observation can be attributed to the fact that, with a greater spatial localization of electron and hole wavefunctions inside the core,

there will be a higher resistance to the electric field effect. In contrast, when the particles are confined to the shell, they exhibit less localization, reducing resistance to wavefunction polarization. The decrease in energies with increasing electric field can be attributed to the strong localization of the wavefunctions of the studied particles associated with the newly added $z$-dependent linear term in the potential energy. For instance, in the case of an electron confined in the ZnTe/CdSe QD, the wavefunction predominantly resides in the lower region of the quantum dot ($z = -R_2$), while the hole wavefunction is localized at $z = R_1$. Similarly, for the CdSe/ZnTe QD, the electron wavefunction is located at $z = -R_1$, while the hole wavefunction is situated in the upper portion of the quantum dot at $z = R_2$.

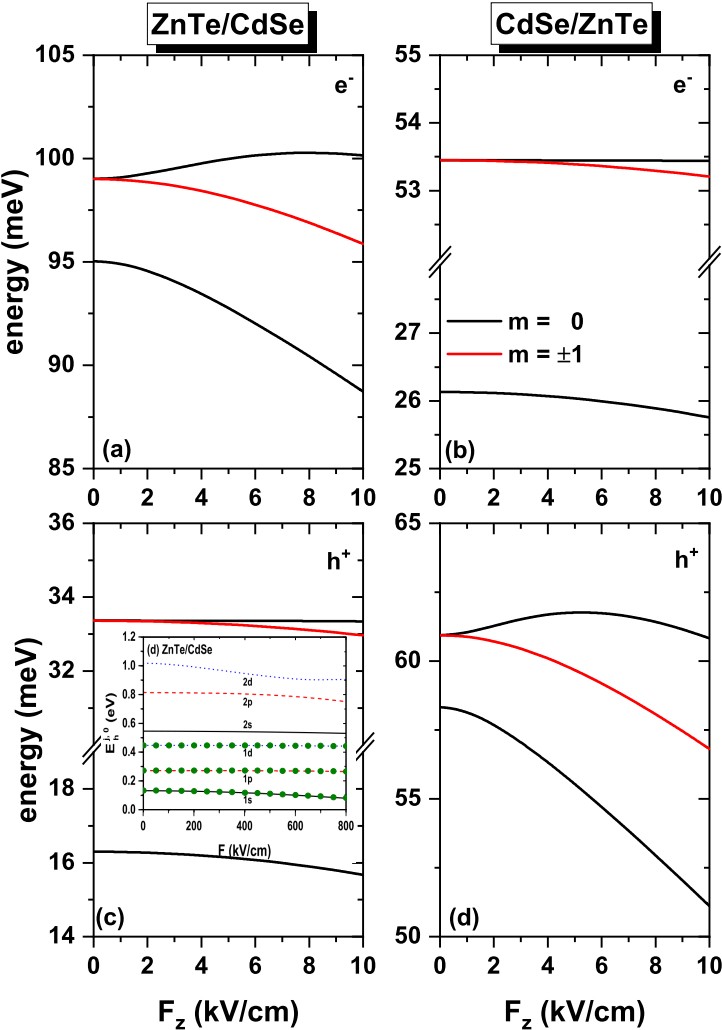

**Figure 4.** Electronic states of a spherical core/shell quantum dot of ZnTe/CdSe (**a,c**) and CdSe/ZnTe (**b,d**) as functions of the applied electric field, $F_z$. Panels (**a–d**) present the results for electrons/holes with $R_1 = 10$ nm and $R_2 = 15$ nm, for different values of the $m$-quantum number, and without magnetic field effects. The inset in panel (**c**) corresponds to the comparison made with the theoretical work of Holovastky et al. [29].

In the inset of Figure 4c, a comparison is shown between the method here used and the theoretical results obtained in 2022 by Holovatsky et al. Their study uses the diagonalization method to solve for a spherical type II core-shell QD of ZnTe/CdSe and CdSe/ZnTe under the influence of an electric field. The same internal radius (3 nm) and external radius (5 nm) parameters used by the authors were employed, and the electric field was varied from 0 to 800 kV/cm. In the inset, the states 1s, 1p, 1d, 2s, 2p, and 2d correspond to the results reported by Holovatsky et al., while the green dots on the 1s, 1p, and 1d curves represent

our results. Based on this comparison, it can be stated that an excellent agreement was achieved between both data sets.

Figure 5 presents the results for exciton energy, calculated using the Coulomb integral described by Equation (10), for the ground state ($m = 0$) of an electron–hole pair in CdSe/ZnTe and ZnTe/CdSe spherical core/shell QDs. In Figure 5a, the exciton energy is plotted as a function of the magnetic field, $B$, while in Figure 5b, it is shown as a function of the electric field, $F_z$. Moreover, Figure 5c displays the exciton energy as a function of the inner radius, $R_1$. In Figure 5a, a slight decrease in exciton energies can be observed as the magnetic field increases. For instance, at zero field, the exciton energy is 7.87 meV, while at 30 T, it is 7.02 meV for the CdSe/ZnTe system and 7.00 meV for the ZnTe/CdSe system. So, there is a variation of approximately 0.87 meV. The decrease in energy values is attributed to the influence of the magnetic field, causing an increase in the expectation value of the electron-hole separation distance, which goes from 9.60 nm at $B = 0$ to 10.84 nm at $B = 30$ T (see inset in Figure 5a). This reflects in a weakening of electron-hole Coulomb interaction. On the other hand, Figure 5b illustrates the exciton energy as a function of the external electric field, applied along the $z$-direction ($F_z$). The figure shows a decrease in exciton energies with the presence of the electric field, ranging from 7.87 meV (at $F_z = 0$) to 6.67 meV (CdSe/ZnTe) and 6.56 meV (ZnTe/CdSe) for an electric field of 10 kV/cm. This decrease in exciton energy arises because the electric field confines the electron and hole wavefunctions to opposite regions, while the electron wavefunction is redistributed in the lower region of the QD, the hole wavefunction occupies the upper region. This separation of the probability densities leads to an increase in the expectation value of the electron-hole distance, from 9.65 nm ($F_z = 0$) to 11.32 nm (CdSe/ZnTe) and 11.51 nm (ZnTe/CdSe) at $F_z = 10$ kV/cm (see inset in Figure 5b), resulting in a reduction in exciton energy. Figure 5b also indicates that the ZnTe/CdSe system is more affected by the electric field compared to the CdSe/ZnTe system. This behavior is expected since, in the ZnTe/CdSe system, the electron is confined to the shell, where it has a lower effective mass than the hole. Consequently, the electron experiences stronger effects from the electric field than the hole confined in the shell (CdSe/ZnTe system). Figure 5c illustrates the energy of the exciton as a function of the magnetic field with an applied electric field of 5 kV/cm to the system. As expected, the presence of the electric field affects the exciton energy compared to results that solely considers the magnetic field (Figure 5a). Specifically, the exciton energy is reduced by approximately 0.71 meV, as also evident in the graph showing exciton energies as a function of the electric field (Figure 5b). Apart from this difference, the behavior of the exciton energy with respect to the magnetic field remains the same, displaying a decreasing trend as the magnetic field increases. This energy decay is attributed to the increase in the electron-hole distance, which corresponds to the elevated magnetic field (see inset in Figure 5c). In Figure 5d, a decrease in exciton energy can be observed as the inner radius, $R_1$, increases. This reduction is attributed to the increase in the core size with the expansion of $R_1$, leading to a larger distance between the electron and the hole (see inset in Figure 5d). Consequently, the strength of the Coulombic interaction between the electron and hole diminishes. It is worth noting that this phenomenon is similar for both systems, regardless of whether the electron is confined in the core and the hole in the shell (CdSe/ZnTe, represented by the blue curve), or when the electron is confined in the shell and the hole in the core (ZnTe/CdSe, represented by the red curve). This similarity indicates that the decrease in exciton energy, resulting from changes in the internal radius, is not dependent on the type of confining material but rather on the geometry of the system.

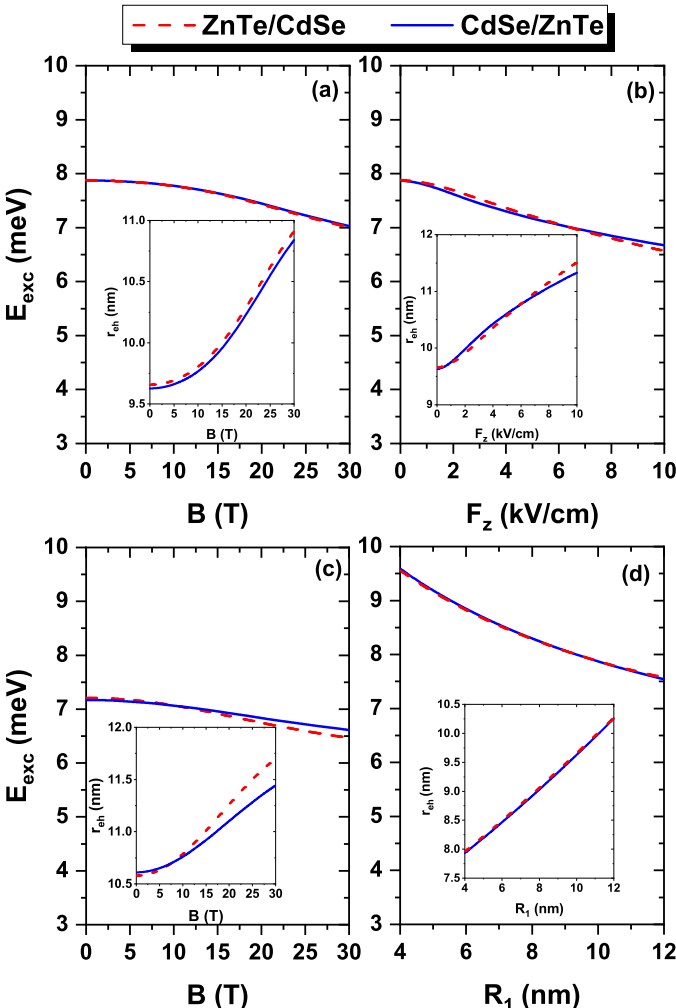

**Figure 5.** Exciton energy as a function of the applied magnetic field (**a**), applied electric field (**b**), (**c**) applied magnetic field with $F_z = 5\,\mathrm{kV/cm}$ (with $R_1 = 10\,\mathrm{nm}$, and $R_2 = 15\,\mathrm{nm}$), and (**d**) internal radius size (with $R_2 = 15\,\mathrm{nm}$). Results are for $m = 0$. The solid-blue (dashed-red) line corresponds to CdSe/ZnTe (ZnTe/CdSe) QD. The inset in each panel shows the corresponding dependence of the expectation value of the electron-hole separation ($r_{eh}$). Panel (**a**) is without an external electric field, panel (**b**) is without an external magnetic field, and panel (**d**) does not present any external field.

Figure 6 shows the energy levels as functions of the magnetic field in the presence of an electric field of 5 kV/cm. The energy curves exhibit similar behavior when only the magnetic field is active, as shown in Figure 3. However, when the magnetic field is set to zero, the excited states display a splitting phenomenon induced by the electric field. For instance, at $B = 0$, the first excited state manifests two distinct energy levels, while the second excited state exhibits three of them, as depicted in the figures with an electric field only (see Figure 4). Notably, these effects are more pronounced when the electron and hole are confined in the shell region (Figure 6a,d), but they are also present when the particle is confined in the core (Figure 6b,c). This distinction is particularly evident for the first excited state, as shown in the inset of Figure 6b, where the energy difference at $B = 0$ is approximately 0.1 meV, compared to around 2 meV in Figure 6a,d.

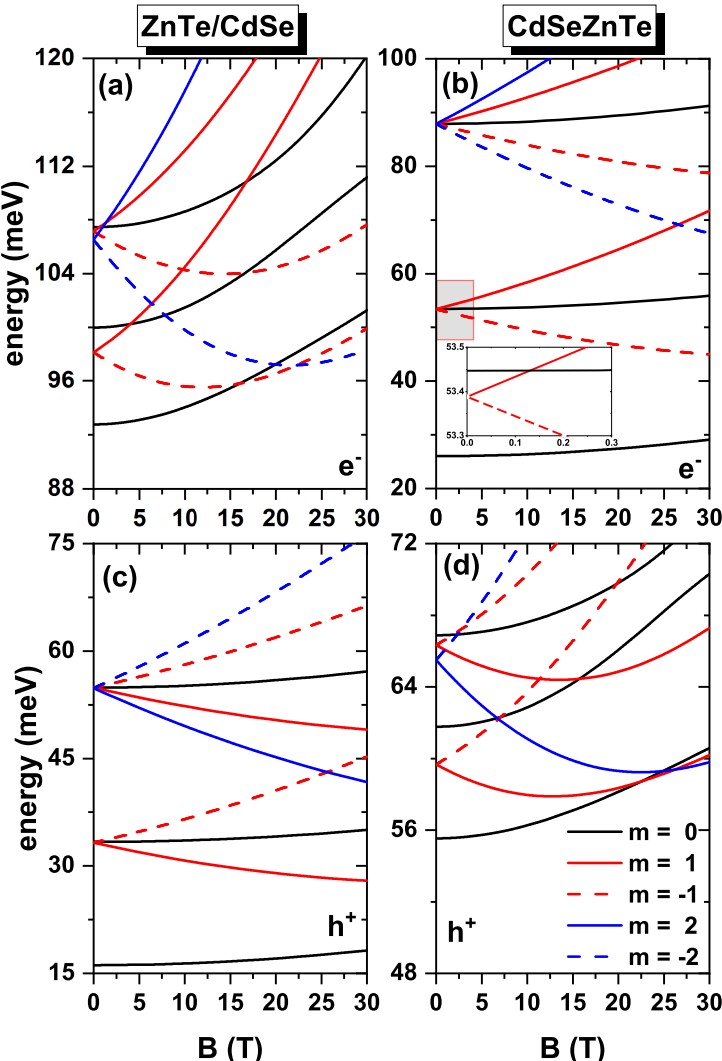

**Figure 6.** Electronic states of a spherical core/shell quantum dot of ZnTe/CdSe (**a**,**c**) and CdSe/ZnTe (**b**,**d**) as functions of the applied magnetic field, *B*, with $F_z = 5 \, \text{kV/cm}$. Panels (**a–d**) present the results for electrons/holes with $R_1 = 10 \, \text{nm}$ and $R_2 = 15 \, \text{nm}$, for different values of the *m*-quantum number.

We present in Figure 7a the intensity of oscillator strength related to interband transitions as a function of the applied magnetic field for CdSe/ZnTe QDs (solid blue curve) and ZnTe/CdSe QDs (red dashed curve). It is observed from the figure that, for both systems, the magnitude of this quantity decreases as the magnetic field strengthens. This is expected since band transitions' oscillator strength is directly proportional to the superposition of electron and hole wavefunctions, which diminishes with the increasing magnetic field. Specifically, when the electron or hole is confined to the shell, their distribution shifts towards the upper and lower regions of the quantum dot as long as the magnetic field intensity augments (compare the first and second column in panel (d) and in the inset in panel (a)), thus resulting in a reduced overlap. In Figure 7b, it is observed that the amplitude of the oscillator strength for interband transitions diminishes as the electric field intensifies. This decrease is evident for CdSe/ZnTe and ZnTe/CdSe systems. The diminishing amplitude of the oscillator strength is attributed to the decreased overlap between the wavefunctions of particles confined to the shell and core. At zero electric field, the wavefunctions of the shell and core particles are confined throughout the spherical region. However, with a non-zero electric field, the hole wavefunction shifts towards the upper region of the quantum dot, while the electron wavefunction shifts towards the lower region (compare the first and third column in panel (d) and in the inset in panel (a)), leading to a reduction in the overlap. Furthermore, in Figure 7c, a decrease in the magnitude of

the oscillator strength is observed in the range from 4.0 nm to 7.6 nm, where its value goes down from 4.69 (at $R_1 = 4$ nm) to 2.64 for the CdSe/ZnTe system and 2.93 for ZnTe/CdSe at $R_1 = 7.6$ nm. However, the opposite trend is observed in the range from 7.6 nm to 12 nm, as the magnitude value of the oscillator strength increases, reaching a maximum value of 5.97 for CdSe/ZnTe and 8.35 for ZnTe/CdSe at $R_1 = 12$ nm. The behavior of the oscillator strength described above can be attributed to the superposition of the electron and hole wavefunctions. The inset of Figure 7c provides insight into this phenomenon by showing the overlap volume of the revolution solid formed by the two wavefunctions at different radii (4 nm, 7.6 nm, and 12 nm). It is clear that, for an internal radius of 4 nm, the revolution solid representing the superposition of the two wavefunctions has a larger volume compared to the case of 7.6 nm. Conversely, the overlap volume is more significant for a radius of 12 nm than in the previous two cases.

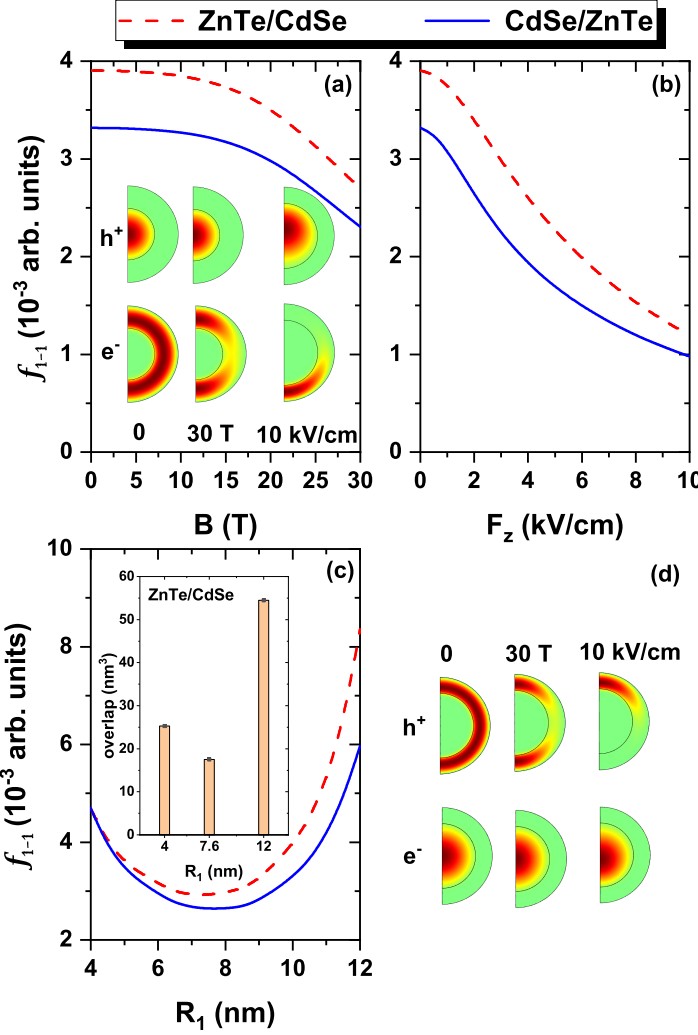

**Figure 7.** Oscillator strength for interband transitions as a function of the applied magnetic field (**a**), applied electric field (**b**) (with $R_1 = 10$ nm, and $R_2 = 15$ nm), and (**c**) the internal radius (with $R_2 = 15$ nm). The solid-blue (dashed-red) curve represents the CdSe/ZnTe (ZnTe/CdSe) QD results. In panel (**d**) are shown the distributions of the ground state wavefunctions for the CdSe/ZnTe QD considering several setups of the applied magnetic and electric field: zero fields (first column), $B = 30$ T with $F_z = 0$ (second column), and $B = 0$ with $F_z = 10$ kV/cm (third column). The identical wavefunction distributions but for ZnTe/CdSe QDS are shown in the inset of panel (**a**). The inset in (**c**) is the value of the volume overlap of the wavefunctions for different $R_1$-radius: 4 nm, 7.6 nm, and 12 nm. Panel (**a**) is without an external electric field, panel (**b**) is without an external magnetic field, and panel (**c**) does not present any external field.

Figure 8 contains the plots of the optical absorption coefficient as a function of the incident photon energy for three different magnetic field values. Figure 8a presents the results for ZnTe/CdSe QDs, while Figure 8b displays the corresponding ones for CdSe/ZnTe QDs. The figures demonstrate that the absorption peaks undergo a blueshift as the magnetic field increases. Additionally, a decrease in the amplitude of the resonant peak can be observed. The displacement of the absorption peaks to higher energies can be attributed to the increase in transition energy as the magnetic field strength rises (see the inset in Figure 8a). The oscillator strength influences the amplitude of the absorption peaks (see Figure 7a), which decreases with the magnetic field, thus impacting the overall magnitude of absorption. As noticed from Equation (9), the magnitude of absorption response is directly proportional to the oscillator strength. Furthermore, a distinction can be noted in the peaks and amplitudes between the two studied systems. For instance, in the ZnTe/CdSe system, at a magnetic field intensity of 30 T, the absorption peak is observed at an energy of 1.04 eV, whereas in the CdSe/ZnTe system, the peak appears at 0.99 eV. This difference in energy and magnitude, with ZnTe/CdSe exhibiting a higher amplitude than CdSe/ZnTe, can be attributed to the difference in the dependence of transition energy and oscillator strength between the two systems (see the inset in Figure 8a and the overall behavior in Figure 7a).

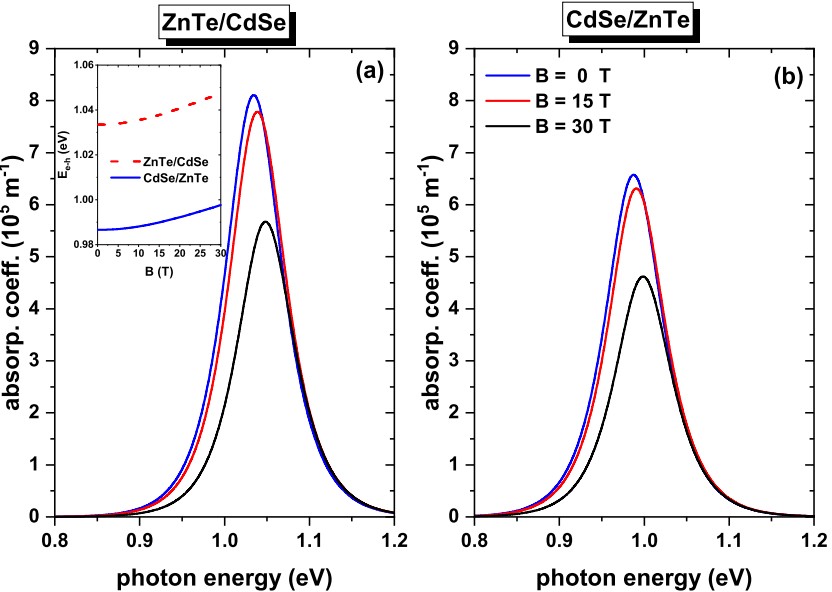

**Figure 8.** Light absorption coefficient as a function of the energy of the incident photon for three different magnetic fields and zero electric fields: $B = 0$ (blue), 15 T (red), and 30 T (black). In (**a**), the results are for ZnTe/CdSe QDs, whereas in (**b**) are for CdSe/ZnTe QDs. The inset in panel (**a**) corresponds to the transition energy as a function of $B$: red-dashed line for ZnTe/CdSe QDs and blue-solid one for CdSe/ZnTe. Calculations are for $R_1 = 10$ nm, $R_2 = 15$ nm, and without an external electric field.

Figure 9 depicts the absorption coefficient as a function of the incident photon energy for three different electric field values in the absence of an external magnetic field. Figure 9a presents the results for ZnTe/CdSe QDs, while Figure 9b displays the results for CdSe/ZnTe QDs. From both figures, a notable change observed upon introducing the electric field is the decrease in the amplitude of the absorption coefficients. This decrease is directly related to the oscillator strength (see Figure 7b), as it diminishes rapidly with an increasing electric field. Furthermore, the figures show that the absorption peaks undergo a slight shift towards lower energies for both systems. This shift is associated with the transition energy (see the inset in Figure 9a), which decreases with the electric field. However, the decrease is approximately 6 meV for both systems. It is important to mention that in the case of the ZnTe/CdSe system, the absorption peaks are located at higher energies of the incident

photon and exhibit a greater amplitude compared to the CdSe/ZnTe system. This disparity arises from the fact that the ZnTe/CdSe QDs have higher transition energy and oscillator strength amplitudes (see Figure 7b).

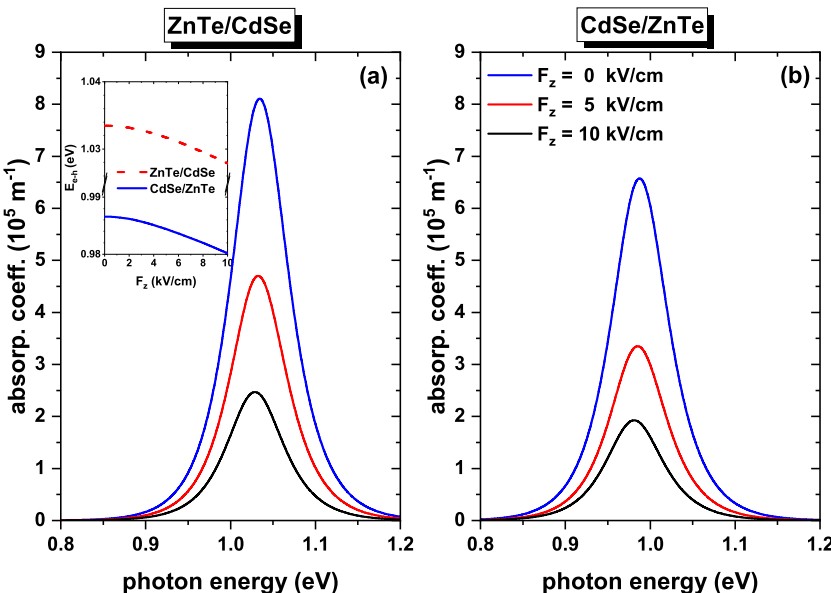

**Figure 9.** Light absorption coefficient as a function of the energy of the incident photon for three different electric fields and zero magnetic fields: $F_z = 0$ (blue), 5 kV/cm (red) , and 10 kV/cm (black). In (**a**), the results are for ZnTe/CdSe QDs, whereas in (**b**) are for CdSe/ZnTe QDs. The inset in panel (**a**) corresponds to the transition energy as a function of $F_z$: red-dashed line for ZnTe/CdSe QDs and blue-solid one for CdSe/ZnTe. Calculations are for $R_1 = 10$ nm, $R_2 = 15$ nm, and without external magnetic field.

Figure 10 illustrates the absorption coefficient as a function of the incident photon energy for three different values of the internal radius in the absence of external fields. Figure 10a displays the ZnTe/CdSe QDs results. It can be observed that the peaks at 4 nm and 7.6 nm exhibit similar amplitudes but at different incident photon energies. Specifically, one peak is located at 0.99 eV (7.6 nm), while the other is slightly shifted to the right at 1.03 eV (4 nm). In the case of the internal radius of 12 nm, the peak displays a greater amplitude than the previous cases, and its energy is blueshifted to 1.17 eV. The observed behavior in the energy position of the absorption peaks is directly related to that of the transition energies (see the inset in Figure 10a). The transition energies show a minimum value at 7.6 nm, an intermediate value at 4 nm, and a maximum value at 12 nm. This variation occurs because the energy of the hole ground state (see Figure 2a) and the energy of the exciton (see Figure 5c) decrease more rapidly in the smaller radius values than the energy of the electron increases. However, above 8 nm, the energy of the electron increases at a faster rate. Regarding the difference in amplitude, it can be related to the oscillator strength (see Figure 7c), which exhibits a maximum value for $R_1 = 12$ nm. In Figure 10b, the results for CdSe/ZnTe QDs are presented. The absorption coefficient peaks for the 12 nm and 4 nm values of radius occur at similar incident photon energies, specifically at 1.06 eV. However, they differ in amplitude, with the peak at 12 nm displaying a greater amplitude. In the case of $R_1 = 7.6$ nm, the peak is shifted to the left (towards the red region) at 0.97 eV, and it has a smaller amplitude compared to the previous cases. As discussed previously, the positions of the resonant absorption peaks and their amplitudes can be attributed to the corresponding values of transition energy and oscillator strength. For the 12 nm and 4 nm radii, they possess the same transition energy (see the inset in Figure 10a). Regarding the amplitude, the oscillator strength is higher for the 12 nm case, resulting in a greater amplitude for the 4 nm and 7.6 nm cases (see Figure 7c).

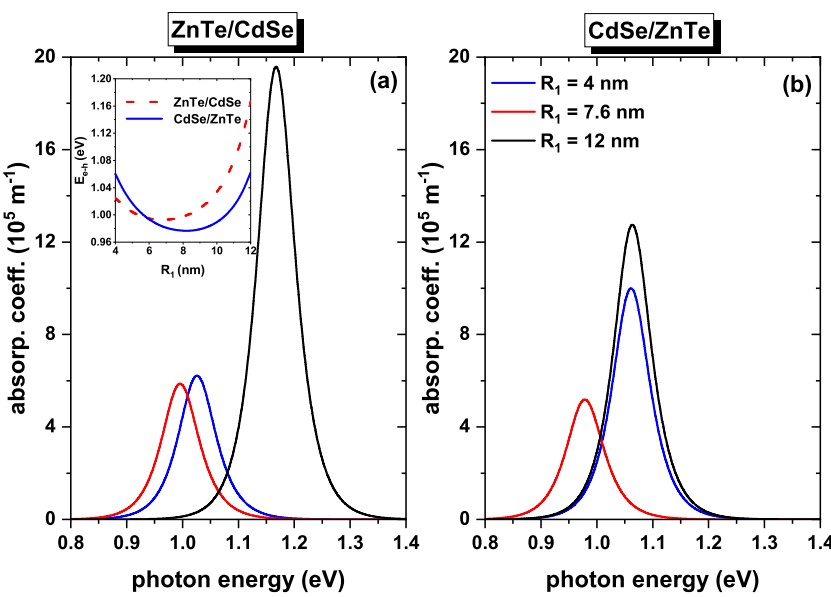

**Figure 10.** Light absorption coefficient as a function of the energy of the incident photon for three different internal radii for zero electric and magnetic fields: $R_1 = 4$ nm (blue), 7.6 nm (red), and 12 nm (black). In (**a**), the results are for ZnTe/CdSe QDs, whereas in (**b**) are for CdSe/ZnTe QDs. The inset in panel (**a**) corresponds to the transition energy as a function of $R_1$: red-dashed line for ZnTe/CdSe QDs and blue-solid one for CdSe/ZnTe. Calculations are for $R_2 = 15$ nm, and without external field.

Finally, we wish to emphasize that our results represent a phenomenological study of the electron, hole, and electron–hole interacting pair confined in type-II spherical core-shell heterostructures. We primarily focus on presenting the basic aspects of the problem and do not claim to provide a direct comparison with experimental results. It is important to note that certain additional effects have not been taken into account, such as: *(i)* The image charge effect, which arises from the significant differences in the dielectric constants between the core and shell regions. *(ii)* The influence of radial and tangential stresses and strains associated with the values of lattice constants of the core and shell. When approaching our results, it must be also borne in mind that we have not considered variations in the electron and hole effective masses as the quantum dot (QD) size changes. In a comprehensive and detailed study of confined excitons in spherical QDs with infinite confinement, S. Pokutnyi [46,47] has presented a modified effective mass approximation. He demonstrated that as the size of the QD approaches the dimensions of the effective Bohr radius, an accurate description of the excitonic states necessitates considering the dependence of the effective mass on the QD's radius. These effects could be relevant in providing a more comprehensive understanding of the system's behavior.

To validate our findings against experimental results, we conducted a comparison with the photoluminescence peak obtained by Jiang and Kelley for ZnTe/CdSe core/shell nanocrystals, where the shell was grown using the Stranski–Krastanov technique [48]. According to the authors' report, the ZnTe core has a radius of 1.3 nm, and the CdSe shell has a thickness of 0.9 nm. After an annealing time of 80 min, they achieved a stable structure with an emission peak at 1.669 eV. Jiang and Kelley's work involved comparing their experimental results with a theory that combines the effective mass approximation, interdiffusion at the core/shell interface, and radial and tangential stresses and strains.

In our study, we considered that for ZnTe/CdSe QDs, the electron and hole are located in the shell/core region, with effective Bohr radius and Rydberg energy values of $a_B = 5.57$ nm and $Ry = 15.56$ meV, respectively. Combining the core and shell dimensions from Jiang and Kelley's work, we concluded that the system's dimensions are of the order of ~0.39, $a_B$, indicating a highly confined system. So, Coulomb interaction, calculated via a perturbative approach using Equation (9), proved to be a good approximation.

To account for high quantum confinement, we implemented the modified effective mass theory suggested by S. Pokutnyi [46,47]. Our calculations for the system described in Ref. [48] yielded a photoluminescence peak energy located at 1.6875 eV, with corresponding electron, hole, and electron-hole interaction energies of 0.653 eV, 0.2023 eV, and 0.099 eV, respectively. Based on these results, we can confidently state that our findings are in good phenomenological agreement with the experimental results, and the predictive numerical experiments presented in this research are physically accurate.

## 4. Conclusions

In conclusion, this study investigated the optical transitions between the hole and electron ground states in type-II spherical QDs, specifically ZnTe/CdSe and CdSe/ZnTe core/shell ones. The effects of changes in the internal radius and the application of external electric and magnetic fields were examined. The calculations revealed several key findings. Firstly, the transition energy rises as the magnetic field increases, while the overlap between the electron and hole wavefunctions decreases. Consequently, the absorption coefficient peaks experience a blue shift, accompanied by a decrease in their amplitude. Secondly, variations in the electric field cause a decrease in the transition energy and the overlap of the electron and hole wavefunctions. This results in a slight red shift of the absorption coefficient peaks and a significant reduction in their magnitude. Furthermore, the exciton energy and the oscillator strength decreased with increasing magnetic and electric fields. The former is linked to the expectation value of the electron–hole distance, which increases under the influence of magnetic and electric fields. The latter is associated with decreased electron and hole wavefunctions overlap. Moreover, changes in the internal radius affected the transition energy and oscillator strength. In the ZnTe/CdSe QD, the transition energy and oscillator strength exhibited a minimum value at 7.6 nm, an intermediate value at 4 nm, and a maximum value at 12 nm. A similar trend was observed for the CdSe/ZnTe QD, where the transition energy and oscillator strength reached a minimum at 7.6 nm. However, for the 4 nm and 12 nm radii, they had similar values of transition energy. The behavior of the transition energy was linked to the electron and hole energies as the internal radius changed, while the overlap of the wavefunctions directly influenced the oscillator strength. As a result of the transition energy and oscillator strength behavior, the absorption coefficients exhibited their highest amplitude and energy of incident photons at an internal radius of 12 nm for the ZnTe/CdSe system. Conversely, at an internal radius of 7.6 nm, the absorption coefficients displayed the lowest amplitude and energy of incident photons. In the CdSe/ZnTe system, the absorption coefficient reached its maximum amplitude at 12 nm, shared the same energy of incident photons with 4 nm, and had the lowest amplitude and energy of incident photons at 7.6 nm.

Our study has been validated with experimental results reported in the literature. Considering the simple model we report here, we believe that we have obtained a good agreement between theory and experiment, which generates confidence in our predictive findings.

**Author Contributions:** R.G.T.-N.: conceptualization, methodology, software, formal analysis, investigation, and writing; J.C.L.-G. and J.A.V.: methodology and software; A.L.M.: formal analysis, investigation, supervision, and writing; M.E.M.-R. and C.A.D.: formal analysis and writing. All authors have read and agreed to the published version of the manuscript.

**Funding:** The authors are grateful to the Colombian Agencies: CODI-Universidad de Antioquia (Estrategia de Sostenibilidad de la Universidad de Antioquia and projects "Propiedades magneto-ópticas y óptica no lineal en superredes de Grafeno", "Estudio de propiedades ópticas en sistemas semiconductores de dimensiones nanoscópicas", "Propiedades de transporte, espintrónicas y térmicas en el sistema molecular ZincPorfirina", and "Complejos excitónicos y propiedades de transporte en sistemas nanométricos de semiconductores con simetría axial") and Facultad de Ciencias Exactas y Naturales-Universidad de Antioquia (ALM and CAD exclusive dedication projects 2022–2023). MEMR acknowledges Mexican CONACYT for support through Grant A1-S-8218.

**Data Availability Statement:** No new data were created or analyzed in this study. Data sharing is not applicable to this article.

**Conflicts of Interest:** The authors declare no conflict of interest.

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
