# Peer review of "Effect of External Fields on the Electronic and Optical Properties in ZnTe/CdSe and CdSe/ZnTe Spherical Quantum Dots"

_condensedmatter, doi:10.3390/condmat8030066_

Round 1

Reviewer 1 Report

Using a finite element method, the authors solved the Schrödinger equation for the electron and hole of spherical Type-II core-shell quantum dots, ZnTe/CdSe and CdSe/ZnTe in a numerically approach. Several approximations were applied in the numerical solution. The obtained wavefunctions were used to estimate the electronic and optical properties of the quantum dots. The influence of external electric field and magnetic field was simulated, and some shifts in the electronic and optical properties were noted.

This work proposed a simple and effective approach to study the electronic and optical properties of quantum dots, which are in the size range smaller than bulk but larger than atomic clusters. In this size range, it is often infeasible to use the conventional computational methods like first-principles calculations and Newtonian mechanics. The proposed method is useful for the specific problem.

This manuscript can be improved in two aspects:

(1)   In introduction, the experimental and theoretical studies on the Typy-II QDs in presence of external fields should be surveyed.

(2)   Lack of comparison with experiments is a serious weakness. The authors did not validate their computational model, their solutions to the Schrödinger equation, their predictions on the electronic and optical properties.  

generally good

Author Response

Referee 1

The Referee:

Using a finite element method, the authors solved the Schrödinger equation for the electron and hole of spherical Type-II core-shell quantum dots, ZnTe/CdSe and CdSe/ZnTe in a numerically approach. Several approximations were applied in the numerical solution. The obtained wavefunctions were used to estimate the electronic and optical properties of the quantum dots. The influence of external electric field and magnetic field was simulated, and some shifts in the electronic and optical properties were noted.

This work proposed a simple and effective approach to study the electronic and optical properties of quantum dots, which are in the size range smaller than bulk but larger than atomic clusters. In this size range, it is often infeasible to use the conventional computational methods like first-principles calculations and Newtonian mechanics. The proposed method is useful for the specific problem.

This manuscript can be improved in two aspects:

Our reply:

The authors want to thank the Referee for his/her comments, which in our opinion have been very pertinent and have helped us to significantly improve the quality of our manuscript.

The Referee:

(1) In introduction, the experimental and theoretical studies on the Typy-II QDs in presence of external fields should be surveyed.

Our reply:

The authors would like to thank the Referee for his/her suggestion. In the introduction section, fifth paragraph, we have added the following text with its corresponding references:

Type II QDs exhibit significant changes in their electronic and optical properties when subjected to external fields. As reported by Holovatsky \textit{et al.} in 2023 \cite{Holovatsky2023}, they conducted a theoretical study on the effects of a magnetic field on the electronic properties and optical quantum transitions between bands in type II ZnTe/CdSe and CdSe/ZnTe QDs. They found that the magnetic field breaks the spherical symmetry of the system, leading to energy spectrum degeneracy concerning the magnetic quantum number. They also observed a dependence of the oscillator strength on the magnetic field, as the field affects the overlap of electron and hole wavefunctions. In 2007, Kuskovsky \textit{et al.} \cite{Kuskovsky2007} presented experimental and theoretical studies on magnetoexcitons in type II QDs formed in Zn-Se-Te multilayers. The aim was to investigate the impact of an intense magnetic field (31\,T) on optical properties. They found that the magneto-photoluminescence exhibited a non-monotonic behavior of the exciton emission intensity as a function of the magnetic field, which manifests the Aharonov-Bohm effect. Their findings were supported by the numerical calculations they performed. Roy \textit{et al.} in 2012 \cite{Roy2012} conducted an experimental study on the influence of magnetic and electric fields on type II ZnTe/ZnSe QDs. They reported robust and narrow Aharonov-Bohm oscillations in the magneto photoluminescence intensity of the stacked QDs due to an embedded electric field. Additionally, they observed a decrease in Aharonov-Bohm oscillations attributed to the electric field.

The Referee:

(2) Lack of comparison with experiments is a serious weakness. The authors did not validate their computational model, their solutions to the Schrödinger equation, their predictions on the electronic and optical properties.

Our reply:

In the inset of Fig. 4(c) we have added a comparison between the results we obtain in our study using the finite element method and the results reported by Holovastky \textit{et al.} [32] for a geometry similar but using a diagonalization method with Bessel functions for the expansion of the wave functions. We have added the following corresponding text to the inset:

In the inset of Fig. \ref{F4}(c), a comparison is shown between the method used in this work and the theoretical results obtained in 2022 by Holovatsky et al. Their study considered a spherical type II core-shell quantum dot of ZnTe/CdSe and CdSe/ZnTe under the influence of an electric field, which was solved using the diagonalization method. For this comparison, the same internal radius (3\,nm) and external radius (5\,nm) parameters used by the authors were employed, and the electric field was varied from 0 to 800\,kV/cm. In the inset of Fig. \ref{F4}(c), the states $1s$, $1p$, $1d$, $2s$, $2p$, and $2d$ correspond to the results reported by Holovatsky et al., while the green dots on the $1s$, $1p$, and $1d$ curves represent our results. Based on this comparison, it can be stated that an excellent agreement was achieved between both data sets.

Reviewer 2 Report

Referee’s report

on the paper «Effect of external fields on the electronic and optical properties in ZnTe/CdSe and CdSe/ZnTe spherical quantum dots» by Rafael G. Toscano-Negrette, José C. León-González, Juan A. Vinasco, Alvaro L. Morales, Miguel. E. Mora-Ramos, and Carlos A. Duque.

         The topic of the paper is interesting and could be useful for readers - physicists if properly executed and presented by the authors. I have several important remarks, especially regarding the analytical part, which is main in the paper. They are the following:

1. The authors use the model of position-dependent effective mass of an electron and a hole, which should be correctly reflected in the Hamiltonian of the system. However, the kinetic energy operator in the original Schrödinger equations (1) is written incorrectly, since in this form the momentum operator p does not affect me(h)(r), as it should be regardless of the coordinate system. This incorrectness further appeared in equations (6), which are the main ones for the further research. Since the solution of equations (6) is further performed by the finite element method by the COMSOL-Multiphysics software, I believe that the equations should be presented in a convincing proper analytical form, rather than using COMSOL-Multiphysics as a "black box" of presented digital elements that the reader does not really needs.

2. What are the reasons for choosing the function δ1-1(Ee-h-ω) in the form of a hyperbolic sequence (11)? At the same time, it is π times greater than its analogous function δ (12) from the paper [40] Sahin M. at all. It is even more unclear: why the authors consider Г=30 meV, as in ref. [40], although the completely different system (CdSe/ZnS) is under study. The values Г are obviously selected under the condition that the maximum of the theoretical function δ coincided with the maximum of the experimental absorption coefficient at the edge of the spectrum?! The choice of Г=30 meV is also surprising because me, mh, Eg in ZnTe are almost twice smaller than in ZnS.

3. It is surprising that although the Hamiltonian of the system contains both electric and magnetic fields, the authors described in detail only the limit cases: Fz=0, B=0; Fz=0, B≠0; Fz≠0, B=0, while and the general case Fz≠0, B≠0 - was not considered. For the sake of completeness, the latter case should be studied.

4. To better organize the design of the paper, the Figures like 2-6 should be placed in one lane and not in different ones. In the caption to Figure 4, there is a mistake: instead of "electric" it says "magnetic" field.

Summarizing, I want to note that since the obtained results of the paper are qualitatively understandable and do not contradict fairly obvious physical considerations, with proper and full consideration of the comments made, and after next positive reviews, with the permission of the journal editors, the paper can be recommended for publication in "Condensed Matter".

Author Response

Referee 2

The Referee:

The topic of the paper is interesting and could be useful for readers - physicists if properly executed and presented by the authors. I have several important remarks, especially regarding the analytical part, which is main in the paper. They are the following:

Our reply:

The authors want to thank the Referee for his/her comments, which in our opinion have been very pertinent and have helped us to significantly improve the quality of our manuscript.

The Referee:

  1. The authors use the model of position-dependent effective mass of an electron and a hole, which should be correctly reflected in the Hamiltonian of the system. However, the kinetic energy operator in the original Schrödinger equations (1) is written incorrectly, since in this form the momentum operator p does not affect me(h)(r), as it should be regardless of the coordinate system. This incorrectness further appeared in equations (6), which are the main ones for the further research. Since the solution of equations (6) is further performed by the finite element method by the COMSOL-Multiphysics software, I believe that the equations should be presented in a convincing proper analytical form, rather than using COMSOL-Multiphysics as a "black box" of presented digital elements that the reader does not really needs.

Our reply:

We thank the Referee for his/her suggestion, which will really make the theoretical framework more transparent and allow it to be used by other researchers regardless of the software or numerical method used to solve eigenvalue differential equations. In the revised version of the manuscript, we have broken down Eq. (1) to obtain the Eq. (2) where the spatial dependence of the effective mass is correctly introduced. Finally, we present the Eq. (3) for the 2D differential equation which is solved via the axis-symmetric module in the COMSOL-Multiphysics software.

The Referee:

  1. What are the reasons for choosing the function δ1-1(Ee-h-ℏω) in the form of a hyperbolic sequence (11)? At the same time, it is π times greater than its analogous function δ(12) from the paper [40] Sahin M. at all. It is even more unclear: why the authors consider Г=30 meV, as in ref. [40], although the completely different system (CdSe/ZnS) is under study. The values Г are obviously selected under the condition that the maximum of the theoretical function δ coincided with the maximum of the experimental absorption coefficient at the edge of the spectrum?! The choice of Г=30 meV is also surprising because me, mh, Eg in ZnTe are almost twice smaller than in ZnS.

Our reply:

We thank the Referee for his/her observation. We have corrected Eq. (11) of the original version of the manuscript. In the revised version, we have included the factor $1/\pi$. We want to point out that there are multiple representations of the $\delta$-function, and this is just one of them. We could well have used other representations, without the physics of the problem undergoing any modification.

Regarding the parameter $\hbar\Gamma$, we have used the value of 30\,meV, which is typically reported in the literature for the case of CdSe. When mentioning this value, we have cited two references, they are:

Rodríguez-Magdaleno, K.A.; Pérez-Álvarez, R.; Ungan, F.; Martínez-Orozco, J.C. Strain effect on the intraband absorption coefficient for spherical CdSe/CdS/ZnSe core–shell–shell quantum dots. \textit{Mater. Sci. Semicond. Process.} \textbf{2022}, \textit{141}, 106400.

Şahin, M.; Nizamoglu, S.; Kavruk, A.E.; Demir, H.V. Self-consistent computation of electronic and optical properties of a single exciton in a spherical quantum dot via matrix diagonalization method. \textit{J. Appl. Phys.} \textbf{2009}, \textit{106}, 043704.

The Referee:

  1. It is surprising that although the Hamiltonian of the system contains both electric and magnetic fields, the authors described in detail only the limit cases: Fz=0, B=0; Fz=0, B≠0; Fz≠0, B=0, while and the general case Fz≠0, B≠0 was not considered. For the sake of completeness, the latter case should be studied.

Our reply:

We thank the Referee for his/her suggestion. We are in complete agreement about the need to combine electric and magnetic fields. The Figs. 5(c) and 6 are new in the revised version of the manuscript.

In Fig. 5(c), we present the exciton energy as a function of the applied magnetic field in the presence of an electric field of 5 kV/cm.

In Fig. 6 of the revised version of the manuscript we present the combined effects of electric and magnetic fields on the electron and hole levels in both core/shell type II heterostructure geometries. In particular, the dependence on the magnetic field for a fixed electric field of 5 kV/cm is presented. We have added the following discussion:

Figure \ref{F55} depicts the energy levels as a function of the magnetic field in the presence of an electric field of 5\,kV/cm. The energy curves exhibit similar behavior when only the magnetic field is active, without an electric field (see Fig. \ref{F3}). However, when the magnetic field is set to zero, the excited states show a splitting phenomenon caused by the electric field. For instance, at $B=0$, the first excited state manifests two distinct energy states, while the second excited state displays three states, as shown in the figures with an electric field only (see Fig. \ref{F4}). Notably, these effects are more prominent when the electron and hole are confined in the shell region (Fig. \ref{F55}(a,d)), but if they are present when the particle is confined in the core (Fig. \ref{F55}(b,c)). This distinction is particularly evident for the first excited state in the inset of Fig. \ref{F55}(b), where the energy difference for $B=0$ is approximately 0.1\,meV, compared to around 2\,meV in Figs. \ref{F55}(a,d).

The Referee:

  1. To better organize the design of the paper, the Figures like 2-6 should be placed in one lane and not in different ones. In the caption to Figure 4, there is a mistake: instead of "electric" it says "magnetic" field.

Our reply:

We thank the Referee for his/her comment and proposal regarding the presentation of Figs. 2-6. We have analyzed the situation and found the following: 1) The vertical scales in each of the panels of Figs. 2-6 are, in general, completely different; there is no way to set the same scale for multiple panels within the same figure. The size of the numbers in the format of two rows and two columns is adequate to be displayed without problems, and 3) the information within each figure, reducing its size, would be quite saturated and visually unappealing. We ask the Referee to allow us to keep the format of the figures as it was done in the original version of the article. Fig. 5 of the original version of the article had three panels; in this modified version, it now has four panels. Thus all Figs. 2-6 would be in the same format of two rows and two columns.

The typo in the caption of Fig. 4 has been corrected.

The Referee:

Summarizing, I want to note that since the obtained results of the paper are qualitatively understandable and do not contradict fairly obvious physical considerations, with proper and full consideration of the comments made, and after next positive reviews, with the permission of the journal editors, the paper can be recommended for publication in "Condensed Matter".

Our reply:

Once again, we thank the Referee for his/her observations, comments, and suggestions. We hope that the revised version of our manuscript will be suitable for publication in the journal Condensed Matter.

Reviewer 3 Report

The manuscript “Effect of external fields on the electronic and optical properties in ZnTe/CdSe and CdSe/ZnTe spherical quantum dots” is devoted to theoretical study of electrons and holes confined in core/shell colloidal quantum dots. The studied type II quantum dots are of the great interest nowadays due to possibility of spatial separation of charge carriers required, for instance, in solar cells. The manuscript is devoted to the study of the electron-hole separation caused by cores/shell sizes and applied fields (electric and magnetic).

I have several comments:

1. It is known that splitting of electron (hole) states in the magnetic field is determined by their total angular momentum composed from orbital and spin angular momentum. In the equation 1 authors consider orbital contribution to splitting of electron and hole states in magnetic field. What is the ratio between the linear with respect to B splitting, calculated by authors, and the spin splitting (Zeeman splitting) of electron (hole) states? Is it legit to consider only the orbital contribution?

2. In equation 3, there should be R2 instead of R3.

3. It is known from 1990-th (see for example Ekimov 1993 JOSA paper https://opg.optica.org/josab/abstract.cfm?uri=josab-10-1-100) that in spherical colloidal quantum dots the complex valence band should be considered for an adequate calculation of confined hole states and optical transitions. In the manuscript, authors neglect this fact ant treat holes similarly to electrons as spin ½ particle in the valence band with zero orbital moment. On this reason I have doubts that the proposed theoretical model is adequate for the considered material system.

4.  Authors calculated exciton energy (equation 7) perturbatively. It is correct only if the exciton binding energy is much smaller than all other energies of the system. What is the ratio between the energy of electron and hole gained in magnetic/electric field and the exciton binding energy? If all these energies are comparable, the Coulomb interaction cannot be considered perturbatively.

5. In caption of Fig. 4 electric field instead of magnetic field should be written.

6. Are there any experimental studies which can be explained by the proposed theory? Because present conclusion contains only results of numerical calculations without any discussion how these result can be verified.

Author Response

Referee 3

The Referee:

The manuscript “Effect of external fields on the electronic and optical properties in ZnTe/CdSe and CdSe/ZnTe spherical quantum dots” is devoted to theoretical study of electrons and holes confined in core/shell colloidal quantum dots. The studied type II quantum dots are of the great interest nowadays due to possibility of spatial separation of charge carriers required, for instance, in solar cells. The manuscript is devoted to the study of the electron-hole separation caused by cores/shell sizes and applied fields (electric and magnetic).

I have several comments:

Our reply:

The authors want to thank the Referee for his/her comments, which in our opinion have been very pertinent and have helped us to significantly improve the quality of our manuscript.

The Referee:

  1. It is known that splitting of electron (hole) states in the magnetic field is determined by their total angular momentum composed from orbital and spin angular momentum. In the equation 1 authors consider orbital contribution to splitting of electron and hole states in magnetic field. What is the ratio between the linear with respect to B splitting, calculated by authors, and the spin splitting (Zeeman splitting) of electron (hole) states? Is it legit to consider only the orbital contribution?

Our reply:

We want to thank the Referee for this observation. To answer the Referee's question, we have decided to make a calculation considering the Zeeman effect for at least one of the two heterostructures. Our calculations show that, for example, the ground state for an electron confined in the ZnTe/CdSe quantum dot is split by the Zeeman effect into two states that at 30 T are approximately 0.5 meV apart. In the case of the hole, the corresponding splitting is also 0.5 meV. In the Figure that we present in this report, we show how the ground states of electron and hole unfold in a ZnTe/CdSe quantum dot as a function of the magnetic field considering the Zeeman effect.

The figure is included in the pdf file

In the revised version of the manuscript, in the paragraph that follows Eq. (10), we have added the following comment:

Notably, despite utilizing an external magnetic field, the Zeeman effect has been disregarded to facilitate a more detailed analysis of the energy levels of the particles under investigation because for 30\,T the Zeeman effect is of the order of 0.5\,meV (not shown here); this is very small compared to the orbital splitting in our results.

The Referee:

  1. In equation 3, there should be R2 instead of R3.

Our reply:

We thank the Referee for his/her. The typo in Eq. (3) has been corrected.

The Referee:

  1. It is known from 1990-th (see for example Ekimov 1993 JOSA paper https://opg.optica.org/josab/abstract.cfm?uri=josab-10-1-100) that in spherical colloidal quantum dots the complex valence band should be considered for an adequate calculation of confined hole states and optical transitions. In the manuscript, authors neglect this fact ant treat holes similarly to electrons as spin ½ particle in the valence band with zero orbital moment. On this reason I have doubts that the proposed theoretical model is adequate for the considered material system.

Our reply:

We want to thank the Referee for his observation. We understand his/her concern, which is actually very valid. What would be desirable would be a complete description of the valence band for a much more adequate treatment of hole states. In that sense, a multiband Hamiltonian would be a suitable way to go.

In Ref. “Excitons in spherical quantum dots revisited: Analysis of colloidal nanocrystals. R. L. Restrepo, Walter Ospina, E. Feddi, M. E. Mora-Ramos, J. A. Vinasco, A. L. Morales, and C. A. Duque. Eur. Phys. J.B 93, 109 (pp9) (2020).”, Restrepo and co-authors developed a theoretical model to describe excitonic states in colloidal spherical quantum dots with quasi-infinite confinement potential. Using a two-band model, much simpler than the one reported in our article, they found an excellent match between the theoretical results and the experimentally reported photoluminescence energy, obviously, with minor differences.

This good agreement between theory and experiment with such a simple model has stimulated us to study the phenomenology of type-II quantum dots with the same type of materials in the core and shell regions, considering the effects of electric and magnetic fields. In our study, we are interested in the phenomenology of the problem rather than reproducing some experimental result.

These justifications do not remove us from the responsibility of addressing the problem in greater depth, but that objective goes beyond the scope of this article. In future work, we will consider the Referee's considerations and model a multiband Hamiltonian for our problem. Of course, the multiband problem requires more sophistication in our study. We cannot imagine at this moment how complicated things can be when the effects of the magnetic field are included, a situation that will radically modify all the entries of the matrix Hamiltonian.

In the Introduction section we have added the Reference “Excitons in spherical quantum dots revisited: Analysis of colloidal nanocrystals. R. L. Restrepo, Walter Ospina, E. Feddi, M. E. Mora-Ramos, J. A. Vinasco, A. L. Morales, and C. A. Duque. Eur. Phys. J.B 93, 109 (pp9) (2020)” with the following comment:

Restrepo and co-authors developed a theoretical model to describe excitonic states in colloidal spherical QDs with quasi-infinite confinement potential. Using a two-band model, they found an excellent match between their theoretical findings and the experimentally reported photoluminescence energy, obviously, with minor differences.

The Referee:

  1. Authors calculated exciton energy (equation 7) perturbatively. It is correct only if the exciton binding energy is much smaller than all other energies of the system. What is the ratio between the energy of electron and hole gained in magnetic/electric field and the exciton binding energy? If all these energies are comparable, the Coulomb interaction cannot be considered perturbatively.

Our reply:

We thank the Referee for his/her observation. In our results it can be seen that the energy of the exciton is around 7-8 meV (Coulomb integral), and from figures 3 and 4 it can also be inferred that for the energies with electric and magnetic fields, the difference in electron-hole energy (free particle energies confined in the heterostructure) is greater than 100 meV. Therefore, in our opinion, we consider that the perturbation theory is valid and can be satisfactorily considered.

The Referee in his comment refers to Eq. (7), this is Eq. (9) in the modified version of the manuscript. After the Eq. (9), we have added the following comment with its corresponding references:

We want to highlight that Heyn and co-workers have used this perturbative approach in multiple studies where excitonic states in heterostructures with axial symmetry have been described and found very good agreement with experimental results for the photoluminescence peak energy transition \cite{coul1,coul2,coul3, coul4,coul5}.

The Referee:

  1. In caption of Fig. 4 electric field instead of magnetic field should be written.

Our reply:

The typo has been amended.

The Referee:

  1. Are there any experimental studies which can be explained by the proposed theory? Because present conclusion contains only results of numerical calculations without any discussion how these result can be verified.

Our reply:

In the inset of Fig. 4(c) we have added a comparison between the results we obtain in our study using the finite element method and the results reported by Holovastky \textit{et al.} [32] for a geometry similar but using a diagonalization method with Bessel functions for the expansion of the wave functions. We have added the following corresponding text to the inset:

In the inset of Fig. \ref{F4}(c), a comparison is shown between the method used in this work and the theoretical results obtained in 2022 by Holovatsky et al. Their study considered a spherical type II core-shell quantum dot of ZnTe/CdSe and CdSe/ZnTe under the influence of an electric field, which was solved using the diagonalization method. For this comparison, the same internal radius (3\,nm) and external radius (5\,nm) parameters used by the authors were employed, and the electric field was varied from 0 to 800\,kV/cm. In the inset of Fig. \ref{F4}(c), the states $1s$, $1p$, $1d$, $2s$, $2p$, and $2d$ correspond to the results reported by Holovatsky et al., while the green dots on the $1s$, $1p$, and $1d$ curves represent our results. Based on this comparison, it can be stated that an excellent agreement was achieved be

Round 2

Reviewer 2 Report

The manuscript can be published in present form.

Author Response

We want to thank the Referee for his/her positive opinion about the final version of our manuscript.

Reviewer 3 Report

I think that Authors answered all the questions and significantly improved the manuscript by making corresponding changes in the text. 

Author Response

(The authors gave the same response as above.)
